# Implicit Overhearing Node-Based Multi-Hop Communication Scheme in IoT LoRa Networks

**DOI:** 10.3390/s23083874

**Published:** 2023-04-10

**Authors:** Dick Mugerwa, Youngju Nam, Hyunseok Choi, Yongje Shin, Euisin Lee

**Affiliations:** 1School of Information and Communication Engineering, Chungbuk National University, Cheongju 28644, Republic of Korea; 2Research Institute for Computer and Information Communication, Chungbuk National University, Cheongju 28644, Republic of Korea

**Keywords:** long range (LoRa), low-power wide-area network (LPWAN), Internet of things (IoT), spreading factor (SF), medium access control (MAC), received signal strength indicator (RSSI)

## Abstract

Long range (LoRa) is a low-power wide-area technology because it is eminent for robust long-distance, low-bitrate, and low-power communications in the unlicensed sub-GHz spectrum used for the Internet of things (IoT) networks. Recently, several multi-hop LoRa networks have proposed schemes with explicit relay nodes to partially mitigate the path loss and longer transmission time bottlenecks of the conventional single-hop LoRa by focusing more on coverage expansion. However, they do not consider improving the packet delivery success ratio (PDSR) and the packet reduction ratio (PRR) by using the overhearing technique. Thus, this paper proposes an implicit overhearing node-based multi-hop communication (IOMC) scheme in IoT LoRa networks, which exploits implicit relay nodes for performing the overhearing to promote relay operation while satisfying the duty cycle regulation. In IOMC, implicit relay nodes are selected as overhearing nodes (OHs) among end devices with a low spreading factor (SF) in order to improve PDSR and PRR for distant end devices (EDs). A theoretical framework for designing and determining the OH nodes to execute the relay operations was developed with consideration of the LoRaWAN MAC protocol. Simulation results verify that IOMC significantly increases the probability of successful transmission, performs best in high node density, and is more resilient to poor RSSI than the existing schemes.

## 1. Introduction

The Internet of things (IoT) has profoundly gained popularity as it continues to impact society in numerous application domains, such as industry IoT (IIoT), embraced through tremendous progress in Industrial 4.0. This paradigm shift has not only brought about a swift advancement in communication but has also uplifted the computational systems [1,2]. The ubiquitous property of the IoT is revolutionizing our lives by connecting objects such as smart refrigerators, smart meters, smart vehicles, and smart washing machines [3]. These smart objects have necessitated Internet connectivity with reliable and cost-effective communications to perform various smart functions that require less or no human intervention for resourcefully enabling optimized productivity, fewer human faults, improved quality of production, enriched automation processes, et al.

In industrial automation, the design of reliable and cost-effective IIoT networking solutions should be needed to achieve low latency, high packet delivery ratio, real-time industrial communication, big data analytics, and low power consumption for long-term sustainable production [4].

For this, low-power wide-area networks (LPWANs) have emerged as an IoT backbone to enable low-cost network deployment, constrained power consumption, and a long lifetime for battery-powered devices that sporadically transmit data packets from the sensor field to the sink through wireless communications [5]. Moreover, LPWAN solutions constitute robust modulation and low data rates to attain a long coverage communication range that enables IoT applications to obtain the desired level of performance. For enabling reliable, efficient, and resilient LPWANs in IoT networks, technologies such as LoRa (long range), Sigfox, NB-IoT (narrow band Internet of things), and Ingenu have been developed [6,7]. These technologies usually offer outdoor low-power long-range coverage of more than 15 km in rural areas and 3 miles in an urban area for commercial deployment at a low cost.

Among the technologies of LPWAN, LoRa networking is predominantly deployed in LPWAN applications because it is based on an open-source and unlicensed industrial, scientific, and medical (ISM) sub-GHz band that enables autonomous network set up at a low cost. The property of LoRa makes it compatible with IoT applications such as smart cities, smart farming, and smart homes [8,9]. LoRaWAN is the standard MAC layer for LoRa, which adopts the star of stars topology, as shown in Figure 1. In this topology, end devices communicate with servers on the Internet via gateways to deliver their own data. LoRaWAN MAC exploits ALOHA MAC and time-division multiple access (TDMA) modes to divide the airtime between end devices for communications.

Conventional LoRa networks are single hop, with end devices (EDs) connected to a centralized gateway (GW) through a direct link, which may cause path loss, longer transmission times, and interference with distant EDs with a high spreading factor (SF), thus causing insufficient coverage and an outage of the desired signal in the uplink at the GW. In addition, a signal outage at the GW may arise as a result of the received signal-to-noise ratio (SNR) being below the threshold for each SF required for error-free decoding when signal strength falls below the sensitivity of the receiver and collisions occur as a result of concurrent transmission [10]. Recently, several multi-hop LoRa networks have proposed schemes with explicit relay nodes to partially mitigate the path loss and longer transmission time bottlenecks of the conventional single-hop LoRa by focusing more on coverage expansion. However, existing multi-hop schemes are still susceptible to interference and collisions, which are exacerbated by MAC layer issues, redundancy issues, and inefficient forwarding mechanisms that contribute to subduing performance degradation.

Therefore, this paper proposes an implicit overhearing node-based multi-hop communication scheme named IOMC in IoT LoRa networks. Basically, the IOMC scheme uses implicit relay nodes, also known as opportunistic relay nodes, to perform overhearing in order to promote relay operation. This is accomplished by using overhearing nodes (OHs) with low SFs to improve both the PDSR and the PRR for distant EDs. The data packet is extended to the GW by the best OH node closer to the GW with a low SF, a lower bit error rate (BER), and a higher residual energy. To achieve the desired result, first, IOMC chooses an OH candidate zone to manage the number of OH nodes by creating a circular area known as a “forwarding zone” in which all nodes participate in the process of determining the best OH. To select the best OH node, we first evaluate two parameters: the link reliability using BER and the residual energy. Then, based on the information about the link reliability and residual energy of candidate OH nodes, a ranking system calculates the rank of a candidate OH node $OH_{i}$ in the OH candidate set to determine the best OH node. Finally, the backoff timer is tuned using a backoff-based strategy so that the OH node with the highest rank value is prioritized with a smaller backoff time than the other OH candidate nodes. We compare the performance of IOMC with that of two distinct multi-hop schemes: an implicit e-node scheme [11] with enhanced LoRa and an explicit multi-hop scheme [12] for range extension using relay nodes. For the performance evaluation, we use two metrics: the probability of successful transmission and the number of packets transmitted by a given number of LoRa nodes deployed in a LoRa network. Simulation results conducted in various environments validate that the IOMC scheme achieves a higher probability of successful transmission while reducing the number of transmitted packets more when compared with the two comparison schemes.

The rest of this paper is organized as follows. In Section 2, we examine the related works on the proposed scheme, IOMC. Section 3 provides a detailed description of IOMC. The performance evaluation of IOMC is presented through simulation results in Section 4. Finally, Section 5 entails the conclusion and future work of this paper.

## 2. Related Works

In this section, we provide the related works on the proposed scheme. First, we briefly mention the important characteristics of the specifications of IoT LoRa communications. Next, we review the existing schemes for multi-hop communications in IoT LoRa.

### 2.1. IoT LoRa Communication

LoRa and LoRaWAN are both commonly used for IoT devices, but each has its own niche capabilities and applications. LoRa is a proprietary chirp spreading spectrum modulation scheme developed and patented by Semtech. This enables long-range, low data rate communication over the license-free sub-1 GHz industrial, scientific, and medical (ISM) bands. The effectiveness of LoRa depends on a link budget, which can be modified through changes in code rate (CR), bandwidth (BW), transmission power (Tx), and spreading factor (SF) [11,13], wherein the selection of data rate (from 0.3 kbps to 50 kbps) is a tradeoff between the time on air and the communication range, given that communications with different SFs do not interfere.

**Spreading factor (SF)** is defined as the ratio between the symbol rate and chip rate. LoRa uses different SFs, where SF∈{7,8,…,12} [14]. SF has a great impact on range, sensitivity, and signal-to-noise ratio (SNR), as represented in Table 1. Different SFs enable orthogonal signal transmission, so a receiver can successfully receive distinct signals sent over a given channel at the same time [15]. SF is also referred to as the number of bits that LoRa encodes in a symbol, wherein a chirp using an SF represents $2^{SF}$ bits using a symbol and $M=2^{SF}$ frequencies for a chirp. Therefore, we can also determine the chirp duration, as shown in Equation (Equation 1), where B∈{125,250} kHz.
(1)$$T_{c}=\frac{2^{SF}}{B}$$

**Transmission power (Tx):** A LoRa radio can transmit in the range of [−4, 20] dBm in 1 dB steps. In normal cases, the transmissible ranges are mostly limited to the range of [2, 20] dBm.

**Code rate (CR):** This defines the level of protection against interference. In this case, the forward error correction code (FEC) uses a CR of 4/5 to 4/8 [9,16]. The higher the CR, the more protection against burst interference, and vice versa. The SF bits of information are transmitted per symbol, and the raw bit rate $\left(R_{b}\right)$ is given in Equation (Equation 2).
(2)$$R_{b}=SF\times \frac{BW}{2^{SF}}\times CR$$

**Bandwidth (BW):** LoRa transceivers operate at 125 kHz, 250 kHz, and 500 kHz. Therefore, BW is considered to be a paramount parameter in terms of modulation.

According to [17,18] LoRa currently operates in the sub-GHz ISM band, subjecting its transmission to strict duty cycle (a maximum of 1%) and transmission power regulations. Herein, three regions are defined in which LoRaWANs are expected to operate at a fixed frequency based on local regulations [19]: channel frequencies in MHz for the United States (US902-923), Europe (EU868-870), and Korea (KR920-923).

Nevertheless, the LoRa Alliance built the LoRaWAN protocol on top of LoRa, adding a networking component to it. LoRaWAN network operates over a star of star topology and exploits a mechanism that enables multiple EDs to communicate with the central network server through the gateway(s) [20]. LoRaWAN comprises three types of device classes (A, B, and C) with different capabilities:

**Class A:** This class of LoRaWAN devices has the lowest power consumption. Class A devices spend most of their time in sleep mode [21,22]. In Class A, after the device sends an uplink, it listens for a message from the receive windows RX1 and RX2 after the uplink before going back to sleep, as shown in Figure 2.

**Class B:** Similar to Class A; however, this class opens extra downlink receive windows at scheduled times. End devices also wake up and open a receive window to listen for a downlink according to a configurable, network-defined schedule [23,24]. Periodic beacon signals transmitted by the network allow end devices to synchronize their internal clocks with the network server.

**Class C:** This class supports bi-directional communication with no restriction on the downlink receive window [23]. End devices in Class C mode are used when extremely low power consumption and latency are trivial. Class C end devices implement the same two receive windows as Class A devices, but they do not close the RX2 window until they send the next transmission back to the server [25]. Therefore, they can receive a downlink in the RX2 window at almost any time.

### 2.2. Multi-Hop Communication in IoT LoRa

One of the first attempts to define multi-hop networks using LoRa is LoRaBlink [26]. The authors propose a TDMA protocol that assumes the communication is only between the nodes and a sink and uses beacons to synchronize the nodes in each epoch, thus benefiting from the concurrent transmission feature of LoRa to decode correctly at most one packet in each slot and receiver. Works in [12,27,28] presented a forwarding relay node and clustering approach to not only enhance coverage and concurrent transmission of LoRa networks using multi-hop communication but also make the system more energy efficient. In addition, devices are categorized into several clusters with the motive of streamlining the operations of the network to maximize energy efficiency and consequently prolong the network lifetime. Unfortunately, the forwarding node increases network traffic and redundancy. Authors in [29] presented a novel architecture for a LoRa hybrid that combines over the air activation (OTAA) and activation by personalization (ABP) features to utilize the multi-hop function. In [30,31], authors introduce a multi-hop LoRa linear protocol where every leaf node is capable of transmitting a data packet to the parent node with a different time slot and e2McH, which performs energy-efficient communication in a multi-hop manner over LPWAN, respectively. In [32], the authors propose a minimized latency multi-hop LoRa network protocol for IoT applications. The scheme aims to achieve reliability and low latency when transmitting the data packet. However, the IOMC scheme only overhears and relays data packets from distant nodes with weak transmission power and demodulation failure at the gateway.

Thenceforth, more concerns have also motivated various studies to overcome connectivity drawbacks by having nodes listen to the medium using overhearing techniques. Fundamentally, if the nodes overhear the neighbor’s traffic, they store the packets in their memory for a short period of time for retransmission. Moreover, in [11,33], authors created a study regarding the use of specially programmed e-nodes and cooperative relay nodes, respectively, to perform overhearing. Here, Class C devices act as a transparent range extender with the primary importance of coexisting with other normal nodes, performing overhearing, storing, and forwarding of all overhead packets to the GW. The crux of the matter is that both schemes perform the same service but differ in methodology. In [11], the e-node scheme deploys a single range extender that takes advantage of the network server (NS) capability to deduplicate the same message received from multiple GWs, while a cooperative relay scheme deploys multiple relays. However, these schemes neglect the reliability and energy efficiency of constrained nodes in the IoT LoRa network. Therefore, the IOMC scheme considers a reasonable, energy-efficient OH node selection technique to mitigate the energy challenges.

## 3. Network and System Model

In this section, we first describe an IoT LoRa network model for the proposed scheme (IOMC) and then present a system model based on the network model for IOMC. Generally, the LoRa technology is made of a star of stars topology, in which gateways relay messages between EDs and a central network server. The notations used in this article are summarized in Table 2. This model is characterized by the provision of a long-range and reliable link with a special modulation technique, in which a LoRa GW(s) collects raw data directly from the end devices and forwards it to a network server (NS), which is interconnected by a high-speed backhaul network, typically Ethernet or cellular communications, as shown in Figure 1. The NS is configured to direct messages to appropriate application servers for processing and also provides the necessary services based on the analysis of the collected data. The proposed system model emulates the conventional LoRa architecture by envisaging an implicit multi-hop communication scheme in an industrial IoT environment as a sensor field. We deploy *N* LoRa nodes in Class A and B modes with a Gaussian distribution in the IoT LoRa network, where they periodically transmit data about their measurement to the central gateway (GW) in an interval of *t* seconds. Our system model has no designated relay nodes between LoRa nodes and the GW. We assume that all the LoRa nodes have the same level of transmission power (Tx), and their own spreading factor (SF) is allocated based on their own distance from the GW [34,35]. To avoid the random selection of the transmission channel, our system model adopts a slotted LoRa MAC. Herein, the channel time is divided into slots with a fixed length *T*, and each LoRa node $\left(N_{i}\right)$ with SF (s) enables it to transmit its packet only at the beginning of a slot.

The IOMC scheme is fundamentally based on the overhearing technique, owing to the broadcast nature of wireless channels. Due to this broadcast nature, several nodes in the locality of a sender node may periodically overhear its packet transmissions, even if they are not the intended recipients of these transmissions. For coping with the unsuccessful transmission of a packet from a LoRa node to the GW, the GW at the wake time selects nodes referred to as overhearing nodes (OHs) in order to overhear the packet of the unsuccessful transmission and to retransmit it to the GW while satisfying the duty cycle regulations. Then, the impact of interference in IOMC can be mitigated by limiting the number of retransmissions of unsuccessfully transmitted packets to the GW. IOMC differs from the traditional LoRa communication method where, if an ACK message is not received in the Rx1 and Rx2 short windows, an extra receive window can be opened by the LoRa node. Otherwise, the LoRa node then sets the maximum number of retransmissions to eight and conducts the retransmissions until it receives the ACK message from the GW for the successful transmission. However, this makes the traditional LoRa communication method cause a lot of interference in the network.

In IOMC, an OH node for the retransmission of an overheard packet is selected among a candidate set with a reliable link in terms of bit error rate (BER), residual energy, and low SF ∈{7,8} closer to the GW. Besides the direct channel allocated from a source node to the GW, IOMC also allocates a channel from the source node to the OH node and a channel from the OH node to the GW. Then, the distance parameter for transmissions will be represented as follows: $d_{SGW}$ represents the one-hop distance from the source node to the GW, $d_{SOH}$ represents the one-hop distance from the source node to the OH node, and $d_{OHGW}$ represents the one-hop distance from the OH node to the GW. Furthermore, we consider a set of candidate OH nodes ($OH_{i}$) according to the requirements of a BER performance that is better than $10^{-3}$. Then, the choice of the best OH node in the set reliably depends on the candidate OH node with the best link quality (BER) and energy efficiency to effectively extend the transmission of a packet from a distant source node to the GW.

Figure 3 shows LoRaWAN uplink and downlink transactions. It is important to note that LoRa networks are generally not designed for delay-constrained communication. Figure 3a depicts a regular LoRaWAN message transmission, in which, if the source (S) delivers eight consecutive Class A uplinks to the gateway (GW) without receiving an ACK frame on the RX1 or RX2 receive window, the node will go back to sleep mode until the next time it has data to report. Nevertheless, the source node (S2), which is closer to the GW, transmits an uplink at T(timer) = α−2β−γ seconds and receives an acknowledgment at the receive window RX1, where α represents the retransmission time, β stands for the propagation delay (depends on distance), and γ means the time slot length in one window (depends on the size of the packet). Figure 3b represents the message forwarding mechanism used in the proposed scheme, where a distant source node (S) is seemingly out of coverage from the GW and transmits a Class A uplink to the network server, and, if the packet is not successfully received, the GW manages the periodically listening nodes to select the best OH node among the candidate set to forward the overheard packet at T = $\alpha -\beta -2\gamma -t_{1}$ seconds. Herein, the GW shall reply on the RX1 or RX2 window of the OH node. The OH node will repeat the downlink to S on the RX1/2 windows or an extra RXR window.

## 4. Implicit Overhearing Node-Based Multi-Hop Communication Scheme (IOMC)

In this section, we describe our IOMC scheme in detail. First, we select an OH candidate zone to manage the number of OH nodes appropriately in Section 4.1. Next, we evaluate two parameters, link reliability and residual energy, to select the best OH node in Section 4.2 and Section 4.3, respectively. Then, we determine the best OH node based on the information about the link reliability and the residual energy of candidate OH nodes in Section 4.4. Last, we set up the backoff timer of OH nodes to allow only the best OH node to participate in the task of relaying data from end nodes in Section 4.5.

### 4.1. Selection of OH Candidate Zone

IOMC uses OH nodes to perform the task of relaying data from end nodes to the GW. To do this, IOMC needs to choose the candidate set of OH nodes. However, if the number of OH nodes is large, only some of them participate in relaying, and the rest of them consume energy unnecessarily due to overhearing. Thus, as the initial task, we narrow down the candidate set of OH nodes by using the network information obtained by the GW from all sensor nodes in the sensor field.

As a result, the GW identifies the position of each sensor node using the network information. As shown in Figure 4, we formulate a circular area called a “forwarding zone” (FZ), in which half the distance between the source node (S) and the gateway (GW) is the radius (r) and the middle point between them is the center point (O). Then, the radius (r) is calculated as follows.
(3)$$r=\frac{d_{SGW}}{2}$$
where $d_{SGW}$ is the distance from S to GW and r is the radius of the circle from the center point (O). $d_{SGW}$ is calculated with the position coordinates of S and GW. All the nodes in the forwarding zone (that is, along the circle and within its circumference) can overhear one another and thus participate in the process to determine the best OH. To decide whether a sensor node (i) is located in the forwarding zone, the condition must satisfy Equation (Equation 5), as follows:(4)$${\left(x_{o}-x_{i}\right)}^{2}+{\left(y_{o}-y_{i}\right)}^{2}\leq r^{2}$$
where the position coordinates of the sensor node (i) are $\left(x_{i},y_{i}\right)$ and the position coordinates of the center point (O) are $\left(x_{o},y_{o}\right)$. In IOMC, the forwarding zone provides a boundary area for a set of OH nodes ($OH_{i}$).

In IOMC, the circumference of a circle provides a substantial forwarding zone in which sensor nodes are distributed according to the Poisson point process. When the intensity of LoRa sensor nodes in the forwarding zone is X, and if μ is the node density within the whole of an IoT LoRa network, then the probability that approximately *k* nodes are deployed in the forwarding zone of size A is calculated depending on the transmission range r of nodes, as follows.
(5)$$P\left(k\right)=e^{-\mu .(A)}\frac{{\left(\mu .(A\right))}^{k}}{k!}$$

The probability *P* that at least one sensor node is located within the forwarding zone is represented as Equation (Equation 6).
(6)$$P=1-e^{-\mu .(A)}$$

Consequently, the forwarding zone (FZ) not only truncates out the unnecessary area for relaying but also offers a sizeable set of sensor nodes $FZ=\{N_{1},N_{2},N_{3}\ldots ,N_{m}\}$ from which we select a candidate set of OH nodes $OH=\{OH_{1},OH_{2},OH_{3}\ldots ,OH_{n}\}$, where OH⊆FZ. Firstly, the sensor nodes in the forwarding zone set (FZ) are carefully scrutinized based on the two predetermined metrics, link reliability and residual energy. A sensor node joins an OH candidate set (OH) of OH nodes if and only if its metric values are above a threshold value to contend for the best OH node. Thus, all sensor nodes in the candidate set have the possibility of becoming the best OH node in IOMC.

### 4.2. Evaluation of Link Reliability Using BER

As a metric used for selecting the best OH among sensor nodes in an OH candidate set, IOMC exploits the link reliability of sensor nodes in the forwarding zone set (FZ) because it is one of the important factors for performing the role of the best OH node, which needs to relay data packets from a distant sensor node to the GW. In IOMC, we use BER to evaluate link reliability. The BER of a sensor node in the forwarding zone is computed by the GW because the GW receives data packets from the sensor node periodically. If a sensor node has a BER less than the threshold, it qualifies to join the OH candidate set, and the reverse is true. In consideration of amplify and forward (AF) and decode and forward (DF) relay modes [36,37], the best OH node adopts the decode and forward mechanism to retransmit the received signal to the destination (GW).

According to our basic knowledge of communication theory, BER is based on SNR. Herein, signal-to-noise ratio (SNR) is the ratio between the received signal power and the noise floor power level [38]. Consequently, the higher the SNR, the better the BER. In particular, for an uplink packet from a sensor node *i* to be successfully decoded by the GW, its SNR should be greater than a given SNR threshold with respect to the SF, where the threshold values are provided in Table 1. Therefore, in this study, we analyze the link reliability based on the packet level SNR model. The BER performance of a LoRa modulation is attributed to both Rayleigh and Rician distributions, which are defined as τ and as the SNR for the LoRa communication. τ is expressed as follows:(7)$$\tau =\frac{E_{s}}{N_{0}\times 2^{SF}}$$
where $E_{s}$ depicts the symbol energy, SF is the spreading factor, and $N_{0}$ is the noise value, defined as $N_{0}=-174+10\log_{10}\left(B\right)$. The sensitivities and SNR represented in Table 1 can also be computed given that
(8)$$SNR=P_{Rx}+174-10\;\log_{10}\left(B\right)-NF$$
where $P_{Rx}$ is the receiver sensitivity, B is the bandwidth, and NF is the noise figure and has 6 dB. Authors in [16,39] have made studies on the analysis of BER and coverage performance with both environments of Rayleigh channel fading to ascertain the impact on the link budget for LoRa communication, where the maximal BER is $10^{-3}$ [40]. Thus, the SNR $\gamma_{br}$ of the receiver with its maximum ratio is shown as the following Equation (Equation 9).
(9)$$\gamma_{br}=\gamma_{S,GW}+\sum \frac{\gamma_{S,FN_{i}}\times \gamma_{FN_{i},GW}}{\gamma_{S,FN_{i}}+\gamma_{FN_{i},GW}}$$

In Equation (Equation 9), $\gamma_{S,GW}$ is the SNR for source to GW, $\gamma_{S,FN_{i}}$ is the SNR for source to a node $FN_{i}$ in the forwarding zone, and $\gamma_{FN_{i},GW}$ is the SNR for $FN_{i}$ to GW. $\gamma_{S,GW}$, $\gamma_{S,FN_{i}}$, and $\gamma_{FN_{i},GW}$ are calculated with the following three Equations (Equation 10)–(Equation 12).
(10)$$\gamma_{S,GW}=P_{S}\left|h_{S,GW}\right|{}^{2}/N_{0}$$
(11)$$\gamma_{S,FN_{i}}=P_{S}\left|h_{S,FN_{i}}\right|{}^{2}/N_{0}$$
(12)$$\gamma_{FN_{i},GW}=P_{FN_{i}}\left|h_{FN_{i},GW}\right|{}^{2}/N_{0}$$

In Equations (Equation 10)–(Equation 12), $P_{s}$ and $P_{FN_{i}}$ are the transmission powers of the source node and the forwarding zone node $FN_{i}$, respectively. $h_{S,GW}$ and $h_{S,FN_{i}}$ are the channel coefficients of S to $FN_{i}$ and to GW, while $h_{FN_{i},GW}$ is the channel coefficient of $FN_{i}$ to GW. $N_{0}$ is the Gaussian noise variance coming from the channel.

According to [41,42], the BER is based on the SNR or $E_{b}/N_{0}$ by applying chirp spread spectrum (CSS), and $BER_{CSS}$ is calculated as the Equation (Equation 13).
(13)$$BER_{CSS}=Q\left(\frac{\log_{12}\left(SF\right)}{\sqrt{2}}\frac{E_{b}}{N_{0}}\right)$$

In simple terms, $E_{b}/N_{0}$ is the ratio between the energy per bit and the noise power spectral density and is expressed as the following Equation (Equation 14):(14)$$\frac{E_{b}}{N_{0}}=\frac{P_{s}B}{P_{n}R_{b}\left(S\right)}$$
where $P_{s}/P_{n}$ is the signal-to-noise ratio (SNR), B is the bandwidth, and $R_{b}\left(S\right)$ is the bit rate. Then, when Q() has a value of *z* as the input value, it is calculated as Equation (Equation 15).
(15)$$Q\left(z\right)=1/\sqrt{2\Pi }\int_{z}^{+\infty }e^{\frac{u^{2}}{2}}du$$

To fully understand the concept, we further make a prescriptive assumption that the SFs used for the transmission of different packets are related to distances between the source nodes and the GW. Without loss of generality, we evaluate the BER $P_{e}\left(S,N_{i}\right)$ from the source node to a sensor node $FN_{i}$ in the forwarding zone as well as the BER from $FN_{i}$ to the GW, as shown in Equation (Equation 16).
(16)$$P_{e}\left(S,FN_{i}\right)=Q\left(\frac{\log_{12}\left(SF_{i}\right)}{\sqrt{2}}\times SNR\times \frac{2^{SF_{i}}}{SF_{i}}\right)$$

The instantaneous BERs of the *S*-to-GW, *S*-to-$FN_{i}$, and $FN_{i}$-to-GW channels can be denoted by $\gamma_{S,GW}$, $\gamma_{S,N_{i}}$, and $\gamma_{N_{i},GW}$, respectively. As a result, the BER of the complete channel with an $N_{i}$ node is equal to the sum of $\gamma_{S,FN_{i}}$ and $\gamma_{FN_{i},GW}$. Thus, with a set of $FN_{i}$ nodes, we consider a channel with a minimum BER value to join the OH candidate set. Similarly, the average BER $P_{e}\left(FN_{i}\right)$ of all $FN_{i}$ in the forwarding zone can be computed as Equation (Equation 17).
(17)$$\begin{array}{c} P_{e}\left(FN_{i}\right)=\left(1-P_{e}\left(S,FN_{i}\right)\right)\times P_{e}\left(S,FN_{i},GW\right)+P_{e}\left(S,FN_{i}\right)\times P_{e}\left(S,GW\right) \end{array}$$

### 4.3. Evaluation of Residual Energy

In addition to the link reliability metric for selecting the best OH node, IOMC also adopts the residual energy of sensor nodes. Since they usually have limited energy powered by batteries, the residual energy is also one of the important factors in performing the role of the best OH node. It is actively and implicitly involved in extending the data packet of a distant source node to the GW. As the best OH node, it consumes more energy than other general sensor nodes because it cannot be in the idle (sleeping mode) state with less energy consumption. Accordingly, the residual energy $\left(E_{re}\right)$ of sensor nodes is a substantial determinant for any form of relay to complete data transfer without interruption. For this reason, when a sensor node has residual energy greater than a predetermined energy threshold $E_{th}$, IOMC includes it in the OH candidate set. Thus, a sensor node exchanges the information of its residual energy with the GW for further scrutiny to qualify for an energy-efficient OH candidate set.

Energy efficiency is the most important key requirement to maximize the lifetime of sensor nodes according to the specified duty cycle. Most of the time, IoT LoRa nodes are in sleep mode, unlike the Class C nodes. Thus, communication for transmitting and receiving data packets is a primary factor consuming the energy of IoT LoRa nodes. To evaluate the energy consumption of a single-hop communication for sending a packet, we adopt the energy consumption model in the LoRa computation provided in [43], as follows:(18)$$E_{single}\left(SF,P_{tx}\right)=E_{tx}+E_{rx}$$
where $E_{tx}$ is the energy consumption caused by the transmission, $E_{rx}$ is that caused by the reception, and $P_{tx}$ is the consumed power according to the supply current and supply voltage for the transmitter. Generally, the time on air (ToA) for a packet mainly depends on SF, BW, and CR. Accordingly, the more the packet spends time on the air during transmission, the more nodes deplete their energy. Subsequently, in the multi-hop communication from a source node to the GW through n-hops, the energy consumption to send a packet is computed as follows:(19)$$E_{n-hops}\left(SF,P_{tx}\right)=\sum_{i=1}^{n}\left(E_{tx,i}+E_{rx,i}\right)$$
where $E_{tx,i}$ and $E_{rx,i}$ are the energy consumption of the transmitter and receiver in the i-th hop according to its current configuration on the transmitted power, spreading factor, and bandwidth, respectively.

Upon understanding the energy consumption dynamics of both single- and multiple-hop communications for a LoRa network, we extend our methodology by computing the total consumed energy ($E_{total}$) of a node in the whole network at a particular time (t), the residual energy ($E_{re}\left(t\right)$) of a node in the whole network at a particular time (t), the total residual energy ($E_{total\_re}\left(t\right)$) of all sensor nodes in the forwarding zone at a particular time (t), and the average residual energy ($E_{avg\_re}\left(t\right)$) of the entire sensor nodes in the forwarding zone at a particular time (t).

First, to calculate $E_{total}$, we consider LoRa nodes operating in two main cycles, which are active and inactive states. Then, the total energy consumption $E_{total}$ used by a LoRa node $N_{i}$ at a particular time (t) is given as follows:(20)$$E_{total\_N_{i}}\left(t\right)=E_{inactive\_N_{i}}\left(t\right)+E_{active\_N_{i}}\left(t\right)$$
where $E_{inactive\_N_{i}}\left(t\right)$≃$E_{sleep\_N_{i}}\left(t\right)$ is the energy consumed by the node during its sleep mode and $E_{active\_N_{i}}$ is the dissipated energy by the node during the transmission mode, reception mode, overhearing mode, data measurement, processing, and wake up.

Next, we calculate the residual energy $E_{re\_N_{i}}\left(t\right)$ of a general sensor node $N_{i}$ in an IoT LoRa network at a particular time (t). To do this, it preferentially needs to calculate the energy consumption $E_{s}\left(i\right)$ of a LoRa node in terms of data rate to transmit $X_{packet}$ with a single-hop communication. $E_{s}\left(i\right)$ can be expressed as follows:(21)$$E_{s}\left(i\right)=\frac{X_{packet}}{R_{b}}\times \left(P_{tx}+P_{rx}\right)$$
where $X_{packet}$ is the size of the transmitted packet in bits, $P_{tx}$ and $P_{rx}$ are the energy consumption for a transmit power and receiver power according to the supply current and supply voltage, respectively, and $R_{b}$ is the bit rate defined in Equation (Equation 2). In general, if the node has *n* single-hop communications until a particular time (t), its residual energy $E_{re\_N_{i}}\left(t\right)$ at the time (t) is computed as follows:(22)$$E_{re\_N_{i}}\left(t\right)=E_{0}-E_{sleep\_N_{i}}\left(t\right)-\sum_{j=1}^{n}E_{s}\left(i\right)$$
where $E_{0}$ is the initial energy of the node at the initial time $\left(t_{0}\right)$, $E_{sleep\_N_{i}}\left(t\right)$ is the energy dissipated by the node during inactive mode at the time (t), and $E_{s}\left(i\right)$ is the energy consumption of the single-hop communication.

For a sensor node $FN_{i}$ in the forwarding zone different from a general sensor node $N_{i}$, its total energy consumption $E_{total\_FN_{i}}$ needs to incorporate the overhearing energy that it consumes to overhear and forward $X_{packet}$ of data to the GW with a distance (d) between $FN_{i}$ and the GW. Then, if $FN_{i}$ participates in the operation of overhearing *k* times, and, in the operation of single-hop communication, *n* times, until a particular time (t), $E_{total\_FN_{i}}\left(t\right)$ of $FN_{i}$ at the time (t) is computed as follows:(23)$$E_{total\_FN_{i}}\left(t\right)=E_{sleep\_FN_{i}}\left(t\right)+\sum_{j=1}^{n}E_{s}\left(i\right)+k\times E_{ov}$$
where $E_{ov}$ is the energy consumed during performing the overhearing operation. Then, the residual energy $E_{re\_FN_{i}}\left(t\right)$ of $FN_{i}$ at the time (t) is expressed as Equation (Equation 24).
(24)$$E_{re\_FN_{i}}\left(t\right)=E_{0}-E_{total\_FN_{i}}\left(t\right)$$

Among sensor nodes in the forwarding zone, to choose the nodes to be included in the set of candidate OH nodes, we determine the energy threshold $\left(E_{th}\right)$ by using $E_{re\_FN_{i}}\left(t\right)$. Since the GW has the information $E_{re\_FN_{i}}\left(t\right)$ about sensor nodes in the forwarding zone, it decides $\left(E_{th}\right)$ by an effective evaluation process. To obtain $\left(E_{th}\right)$, we subsequently calculate the total residual energy $E_{total\_re}\left(t\right)$ of all sensor nodes $FN_{i}$ in the forwarding zone at a particular time (t) by Equation (Equation 25).
(25)$$E_{total\_re}\left(t\right)=\sum_{i=1}^{N}E_{re\_FN_{i}}\left(t\right)$$
where N is the number of sensor nodes in the forwarding zone. Then, the average residual energy $E_{avg\_re}\left(t\right)$ of all sensor nodes in the forwarding zone at the time (t) is calculated as Equation (Equation 26).
(26)$$E_{avg\_re}\left(t\right)=\frac{E_{re}\left(t\right)}{N}$$

Thus, we can deduce that the energy threshold $E_{th}$ is considered to be the average residual energy $E_{avg\_re}\left(t\right)$ of all sensor nodes in the forwarding zone. Accordingly, if a sensor node $FN_{i}$ in the forwarding zone has $E_{re\_FN_{i}}\left(t\right)$ more than $E_{th}$ at a particular time (t), it is included in the set of the candidate OH nodes and participates in the process to elect the best OH node in the forwarding zone. On the same principle, all sensor nodes that do not satisfy the condition $E_{re\_FN_{i}}\left(t\right)>E_{th}$ are dropped from the OH candidate set.

### 4.4. Selection of the Best OH Node

To determine the OH candidate set to select the best OH node from a distant source node to the GW, IOMC enables the GW to identify the information on the link quality and residual values of sensor nodes in the forwarding zone. Intuitively, reliable relay link quality is a very vital parameter for achieving a good packet delivery ratio in multi-hop communication scenarios, and high residual energy is also an important parameter for fulfilling the burden task of extending the overheard data packets to the GW. Thus, if a sensor node in the forwarding zone has link quality and residual energy greater than both the threshold link quality and the threshold residual energy, it is added to the OH candidate set.

Among the sensor nodes in the OH candidate set, IOMC selects the sensor node with both optimal link quality and residual energy as the best OH node. To do this, we exploit a ranking system to evaluate sensor nodes in the OH candidate set. The ranking system calculates the rank of a candidate OH node $OH_{i}$ in the OH candidate set as Equation (Equation 27).
(27)$$\begin{array}{c} R\left(OH_{i}\right)=\alpha \frac{L_{BER}\left(OH_{i}\right)-L_{BER(min)}}{L_{BER(max)}-L_{BER(min)}}+\left(1-\alpha \right)\frac{E_{OH_{i},re}-E_{OH_{i},re\left(min\right)}}{E_{OH_{i},re\left(max\right)}-E_{OH_{i},re\left(min\right)}} \end{array}$$
where $L_{BER}\left(OH_{i}\right)$ is the link quality of $OH_{i}$ on bit error rate and is equal to $P_{e}\left(i\right)$, $E_{OH_{i},re}$, which is the residual energy of $OH_{i}$, and α is a weight value that has a range between 0 and 1 (that is, α∈[0,1]).

For applying the normalized values for both BER and the residual energy, we use $L_{BER(max)}$ and $L_{BER(min)}$ for BER and $E_{OH_{i},re\left(max\right)}$ and $E_{OH_{i},re\left(min\right)}$ for the residual energy in Equation (Equation 27). By the ranking system, when a candidate OH node $OH_{i}$ has the highest rank as $\left(R\left(OH_{i}\right)\right)$ in Equation (Equation 28), it becomes the best OH node.
(28)$$OH^{*}=\underset{i\epsilon N}{arg max}\left(R\left(OH_{i}\right)\right)$$
where $OH^{*}$ means the candidate OH node with the highest rank, which becomes the best OH node to implicitly transmit data from the source node to the GW. The rest of the candidate OH nodes serve in the next cycle according to their own ranking order if the best OH node has a situation that fails to support relaying. In other words, in case of any failure of the best OH node (that is, the one with the highest rank) to overhear the data packet of the source node, the timer for the second-ranked candidate OH node using its rank value will time out so that it can overhear and extend the overheard data packet to the GW while satisfying the duty cycle regulations. Thus, upon a successful transmission, an acknowledgement (ACK) message is sent from the GW, then, the rest of the remaining nodes in the candidate set cancel their timers after overhearing the ACK message. Algorithm 1 shows how to select the best OH node. Line 8 analyzes the two parameters, link reliability using BER and residual energy for all listening nodes in the forwarding zone, to produce a candidate set. In this setting, weights are allocated to each metric. In addition, in lines 9 and 10, the ranking algorithm determines the rank of the candidate OH node to select the best OH node to be inserted in the candidate set. Finally, line 14 returns the best OH node.

**Algorithm 1** Best OH node selection$L_{BER}\left(OH_{i}\right)$: link quality of the $i^{th}$ OH node$E_{OH_{i},re}$: residual energy of nodesLnorm: normalized valueNoL: number of listening nodes$R(OH_{i})$: ranking of link quality and residual energy 1: $OHnodes=\{OH_{1},\ldots ,OH_{i},\ldots ,OH_{l}\}$ 2: candidateset(i)≠0 3: N=0 4: **for **$OH_{i}$∈ OHnodes **do** 5:  **if** $!(is\_listen\left(OH_{i}\right))$ **then** 6:     continue 7:  **end if** 8:  **if** $L_{BER}\left(OH_{i}\right)$ ≤ $L_{th}$ && $E_{OH_{i},re}$ ≥ $E_{th}$ **then** 9:     $R\left(OH_{i}\right)=\alpha \times L_{BER}\left(OH_{i}\right)+\left(1-\alpha \right)\times E_{OH_{i},re}$ 10:     $candidateset\left(i\right).insert(R\left(OH_{i}\right),OH_{i})$ 11:  **end if** 12: **end for** 13:  bestOHnode=candidateset(i).begin() 14: 
return bestOHnode


### 4.5. Backoff Timer of OH Candidate Nodes

IOMC needs to use the retransmission of only one candidate OH node in the OH candidate set to avoid redundant retransmissions of the other candidate OH nodes. Accordingly, as soon as the OH candidate node selected as the best OH node has successfully transmitted an overheard data packet to the GW during its transmit window, the rest of the OH candidate nodes have to halt their retransmission of the overheard data packet. This can be achieved through the use of a backoff-based strategy in which a backoff timer is mapped to all the OH candidate nodes in the OH candidate set according to their own ranks $\left(R\left(OH_{i}\right)\right)$. In simple terms, the backoff timer is tuned in such a way that the best OH node with the highest rank value is prioritized with a smaller backoff time than the other OH candidate nodes.

In our backoff-based strategy, each OH candidate node $OH_{i}$ will start its backoff timer with an initial value $T_{i}$ that is inversely proportional to its own rank $\left(R\left(OH_{i}\right)\right)$, dependent on its link quality and residual energy, and is calculated as Equation (Equation 29).
(29)$$T_{i}=\frac{T_{max}}{R(OH_{i})}+random\left(\frac{1}{T_{max}}\right)$$

Here, $T_{max}$ is a constant representing the maximum backoff time that an OH candidate node will wait to overhear the retransmission of another OH candidate node, depending on the units of both link quality and residual energy and taking the units of time (microseconds). The function random(x) provides a value between 0 and *x* and gives additionally different times to prevent having the same backoff timers when some OH candidate nodes have the same rank coincidentally. This form of our backoff-based strategy eliminates the chances of collision between overheard packets by preventing channel contention and simultaneous attempts for retransmission.

In our backoff-based strategy, the best OH node $OH^{*}$ has the least backoff timer because it has the highest rank compared with the ranks of the other OH candidate nodes. If a source node transmits its packet to the GW, all of its OH candidate nodes overhear the transmission and start their own backoff timers. If the GW transmits an ACK message to the source node, all OH candidates overhear the ACK and finish the overhearing and retransmission process by canceling the backoff timers. However, if the GW does not receive the packet from the source node and transmit an ACK, each of the OH candidate nodes waits to retransmit the overheard packet to the GW until its backoff timer expires to 0. In this situation, $OH^{*}$ has the chance for its backoff timer expires first. Then, it retransmits the overheard packet of the source node to the GW. On receiving the overheard packet, the GW transmits an ACK message in the IoT LoRa network. If the other OH candidate nodes receive the ACK message from the GW, all of them cancel their own backoff timers and finish the overhearing and retransmission process. The source node also recognizes the successful transmission of its packet to the GW by receiving the ACK message. However, when $OH^{*}$ transmits the overheard packet to the GW, if its transmission is a failure, the OH candidate node with the second-highest rank has the chance for its backoff timer to expire to 0. Then, it transmits the overheard packet from the source node to the GW. This process continues until the GW receives the overheard packet and transmits an ACK message to the IoT LoRa network, thus, the packet from the source node is successfully transmitted to the GW.

## 5. Performance Evaluation

In this section, we first describe our simulation environment, model, and performance evaluation metrics. We compare the performance of IOMC with the two existing approaches categorized as implicit multi-hop based on an enhanced LoRaWAN node known as the e-node scheme [11] and explicit multi-hop scheme [28]. Finally, we evaluate the performance of IOMC in comparison to the previous communication schemes through simulation results.

### 5.1. Simulation Environment

This setup is a direct depiction of the LoRa communication system in the industrial IoT, where several sensors are deployed in a sensor area with a harsh propagation environment subject to high path loss that may lead to much shorter communication ranges that contrarily affect the long range propagation achieved in line of sight (LoS) and non-line of sight (NLoS) [44]. To determine the effectiveness of the IOMC scheme, we used the NS-3 simulation environment to model multi-hop in the LoRa network through the overhearing mechanism of selected sensors to extend a data packet to the GW, in contrast to the performance of the cooperative relaying scheme and the multi-hop scheme based on an enhanced LoRaWAN node. The fundamental goal of the environment is to model a network with a large diameter in which we can have a high packet delivery ratio with minimal energy dissipation and jitter. NS-3 provides LoRaPhy, LoRaMAC, LoRaChannel, and mobility mode suitable for large networks anchored on discrete event simulation models of a system that changes to its current state at discrete points in the simulation time [45].

Envisaging a transmission environment, where a signal (measure) is forwarded by a distant node to the LoRa GW within the transmission area, can either be perfectly received and decoded or fail to be delivered to the intended destination due to unprecedented challenges. LoRaWAN networks retransmit data after transmission failure to improve dependability. After a certain number of retransmissions, LoRaWAN drops packets [46,47]. However, the proposed scheme centrally manages and regulates the selection and behavior of a node to overhear the transmission from the source nodes and forward the data packet to the LoRa GW. Consequently, the selected OH node periodically switches between overhearing the sensor’s transmissions during the receive window and forwarding the contents of overheard frames to the GW in a transmit window. This operates within the prescribed duty cycle of the selected sub-band. For efficient delivery of data packets, transmissions from an OH node do not interfere with these sensors with good link quality.

We simulate a network topology containing N stationary LoRa nodes with a Gaussian distribution in concentric circles with a GW located at the center of the sensor area to ensure a maximum coverage range. In this paper, we use an SF between 7 and 12 assumed to be directly proportional to the distance away from the GW, with a varying bit rate that ranges from $0.25\leq R_{b}\leq 5.47$ (kbps) and is also dependent on the SF. The orthogonality of various SFs accompanied by simultaneous multiple channel reception in a packet forwarder is ensured. Theoretically, LoRa permits a maximum of $N_{SFmax}$ × $N_{CHmax}=48$ different unconfirmed communications to occur simultaneously [4,48].

Table 3 shows the simulation parameters used in our simulation model. We considered a circular network region with a cell radius of R=7.5 km. Each node transmits a packet of size 30 bytes, on average t=30 (s) per 24 h. Each packet was subjected to more than 1000 simulations, with each simulation lasting 3600 s. The estimated transmit power is 14 dBm for all the nodes, the channel bandwidth is 125≤BW≤500 kHz, and the channel code rate is 4/5, protected by a cyclic redundancy check (CRC) with a 1% duty cycle restriction [17,49].

All the nodes except the GW have the same communication requirements, where each node generates packets and periodically transmits, and every device maintains a duty cycle of 1% of the minimum allowed duty cycle among the considered sub-bands. The distribution of different SFs was done uniformly depending on the distance from the base station, and all the nodes were placed in the radio coverage area of the GW.

For comparison purposes, we consider two existing schemes to discover the differences between them and also ascertain the relevance of our proposed scheme based on the performance metrics, which are categorized into either implicit or explicit multi-hop communication. In [11] is an implicit multi-hop communication that deploys a programmed e-node, also known as a range extender, based on an enhanced LoRaWAN node for industrial IoT, where every time a new uplink is captured, the e-node replicates it and forwards it to the GW to extend the coverage of messages transmitted using a sequence of highest data rates. The e-node is transparent concerning the existing infrastructure to improve the link quality of poorly connected nodes.

Secondly, there is explicit multi-hop communication. In [28], the authors motivate the use of multi-hop LoRa topology limited to a two-hop range extension for an intrinsic demand of smart city applications to evaluate packet reception ratio and energy for 7, 9, and 12 SFs with respect to distance from the GW. Herein, two topologies were presented: first, based on introducing a forwarding relay node and clustering devices with the view of forming a star of stars topology. The authors, however, took no notice of the significance of considering both the number of relays and relay nodes placement in the network since they have a direct impact on packet delivery ratio, range extension, and power reduction. Therefore, the network is said to be more energy efficient only if the relays are optimally placed.

The metrics used for performance evaluation of the proposed scheme and the existing schemes [11,12] are probability of successful transmission and the number of packets. In simple terms,

Probability of Successful TransmissionFrame transmission is considered successful when the frame is transmitted without collision and all the bits of the frame are precisely decoded despite the interference. Considering the harsh environment and capture effects, not all data sent from LoRa nodes can be transmitted to the GW successfully. In the LoRa network, nodes can either transmit confirmed or unconfirmed messages, that is, without downlink messages and acknowledgments, respectively. Therefore, it is important to critically consider the probability for the GW to successfully receive an uplink from the source node and the probability that the node also successfully receives a downlink. Certainly, the proposed scheme is tailor-made to enhance the probability of successful transmission under the condition that the OH node (re)transmits a frame while satisfying the duty cycle regulations. Essentially, the OH nodes only transmit data packets after the unsuccessful transmission of the source node. That is, we consider different SFs, node density in a particular SF, SNR, BER, and duty cycle to evaluate the transmission success probability. We derive the probability of successful transmission $P_{suc}$ of a frame as in Equation (Equation 30).
(30)$$p_{suc(i)}=e^{-2\times N_{sf(i)}\times D}\;\;\;with\;\;0\leq P_{suc}\leq 1$$
where $N_{sf(i)}$ is the density of nodes within a given SF(i)∈{7∼12} and D represents the duty cycle. Therefore, to establish the number of nodes within a specific SF, we deployed the probability density function $PDF=\frac{1}{2}\left(1+erf\left(\frac{x-\mu }{\delta \sqrt{2}}\right)\right),$ where μ is mean, δ is variance, and the error function, also known as the Gauss Error Function, is denoted by erf.Number of PacketsThe number of packets traversing the network to the GW is a vital parameter as far as understanding traffic behavior for either congested or uncongested communication in the LoRa network is concerned. In the LoRaWAN network, packets are transmitted sporadically and depend on many factors. Therefore, the goal here is to ensure an effective transmission without duplicate packets and retransmissions, which create traffic in the network and thus hinder the goodput of the entire network as a result of increased redundancy of packets. Certainly, it is important to note that the number of nodes in a LoRa network is inversely proportional to the throughput.

### 5.2. Simulation Results

In this section, we present the obtained results of the performance evaluation metrics using LoRa nodes’ communication transmission parameters in Table 3, and we compare the performance of the IOMC scheme against the two distinct multi-hop comparing schemes: an implicit e-node scheme with an enhanced LoRa and an explicit multi-hop scheme for range extension using relay nodes. We focused on two major aspects: the probability of successful transmission and the number of packets transmitted by a given number of LoRa nodes deployed in a LoRa network, defined as the node density and the distance between the LoRa nodes and the GW, represented as the average distance and the link quality in terms of RSSI (dBm) values.

This research validates the effectiveness of a proposed centralized implicit scheme for multi-hop LoRa networks by taking advantage of overhearing node(s) (OH) to achieve high levels of data delivery while considering packet retransmission to cope with frame losses. This is contrary to the existing phenomenon of conventional LoRa, where the distribution of successfully transmitted packets from distantly deployed LoRa node(s) is transmitted directly to the GW in one hop at the expense of sending each packet multiple times (eight times) until the data packets are successfully delivered.

When the best OH node is selected, the IOMC eliminates the collision of OH candidate nodes; the probability of two or more OH node timers expiring at the same time is zero. The maximum time an OH candidate node will wait to overhear another OH candidate node’s retransmission is $T_{max}\approx 500\;\mu$s [37]. Moreover, the probability of having two or more relay timers expire within the same time interval c is nonzero, with $T_{max}\approx n\times T_{delay}$, n≈5 (number of candidate OH nodes), and $T_{delay}\approx 100\;\mu$s (time delay for each OH node). Therefore, the range of the random time interval is [0,500μs].

Figure 5 shows the variations in node density with the probability of success and number of packets of the three LoRa communication schemes investigated in this study. The results in Figure 5a show a decrease in the probability of success with increased node density for the e-node and multi-hop schemes. This decrease could be attributed to the ALOHA media access strategy, which causes collisions derived from the blind transmission strategy, as the nodes are allowed to transmit packets as soon as they become ready for transmission [50,51]. However, the IOMC scheme showed no change when 100 nodes were deployed in the network and a significant increase in success probability of approximately 96% with an increasing node density of up to 200 nodes. This could be attributed to the synchronization services and the use of the overhearing strategy. Herein, the OH nodes retransmit messages from previously failed transmissions within the communication range to alleviate multiple retransmission.

There is a direct proportionality between the number of packets transmitted and the node density of all schemes Figure 5b. That is to say, the number of packets increases with an increase in the number of nodes. In the total number of packets transmitted for a duty cycle of 1%, we observe that the IOMC scheme initially registered a relatively low number of packets against the node density within the first 100 nodes and thereafter a slight increase in the number of packets, surpassing the e-node scheme and multi-hop scheme. Therefore, having a sizeable number of OH nodes with fairly good link quality offers efficient retransmission in the event of transmission failure at the source node. Consequently, a gradual rise in the number of packets as the number of nodes increases is justified. Contrary to this, the e-node scheme and multi-hop scheme at this point are susceptible to poor link quality, collisions, and channel contention, resulting in a drop in the number of packets transmitted.

Figure 6 shows the variations in the average distance with the probability of success and the number of packets. In general, there is a decrease in the probability of success with increasing average distances for all the schemes, as shown in Figure 6a. This is attributed to a high probability of collision and capture effect, where messages sent by LoRa nodes that are furthest from the GW experience the highest packet loss ratio (PLR) and longest time on air (ToA), and nodes closest to the GW can choose among multiple low SF values. Therefore, it is likely that the furthest ones can only select the highest SFs for transmission [17]. The noted increase in the multi-hop scheme within the average distances of 6000 to 10,500 (m) could be brought about by the availability of relays that play a fundamental role in successfully delivering messages to the final destination, which also corresponds to [50]. Whereas, in the region between 10,500 and 15,000 (m), there is a performance degradation as a result of a poor link, which does not necessitate an efficient frame transmission to both the relay and GW, similar to the e-node scheme. The findings further show that the IOMC scheme exhibits the best performance over 89% along with various average distances from the GW. This could be due to the significantly increased number of OH nodes that retransmit all the failed transmissions to the GW; this bridges the gap of unsuccessful transmissions.

In Figure 6b, the IOMC scheme has a relatively stable and low amount of packets transmitted compared to the e-node and the multi-hop schemes. Precisely, the e-node scheme exhibits the highest number of packets as a result of the implementation mechanism where the e-node replicates all packets transmitted by source nodes and forwards them to the destination. However, as the average distance increases, the e-node scheme depicts an ameliorated performance: unlike the multi-hop scheme, the number of packets increases as the average distance increases. This occurs as a result of deployed relays located closer to the distant nodes extending data packets from the source to the GW.

Figure 7a shows variations in the link quality represented by RSSI (dBm) with the probability of success and number of packets of the three LoRa communication schemes under investigation. The results show a steady, successful transmission with RSSI values between −100 and approximately −115 (dBm) for all three schemes. As the link continues to fade, the e-node scheme and the multi-hop scheme’s probability of success gradually decreases as a result of the wireless signals being affected by the increase in distance away from the GW and other environmental conditions such as ambient noise and obstacles (e.g., buildings and foliage) [52].

However, the IOMC scheme showed a steadily high performance until the threshold RSSI value of −120 (dBm). Beyond this point, as the RSSI values deteriorated, the probability of success for the IOMC scheme slightly decreased, but not to the extent of the e-node and multi-hop schemes. This could be attributed to the selection strategy of OH nodes with better link quality to retransmit failed messages. In addition, comparing the probability of success of a multi-hop scheme and an e-node scheme with a high node density, the multi-hop scheme achieves a better result in terms of deploying relays to provide substantial benefits in forwarding packets to the GW. Contrary, the e-node scheme’s poor link quality leads to a path loss to the source node, which obviously affects the overhearing of the e-node statically located in the sensor field.

The IOMC scheme outperforms both the e-node scheme and the multi-hop scheme in terms of the reduced number of packets transmitted, as shown in Figure 7b for RSSI values between −100 and −120 dBm. However, we observe an important twist at a point threshold value of −120 dBm, where the IOMC scheme invokes the OH node to retransmit the lost frames as a result of path loss due to the deteriorating link quality. Consequently, a significant increase in the number of packets was observed, contrary to the e-node and multi-hop schemes, whose packets diminished as a result of the failure in transmission due to the poor link quality.

## 6. Conclusions and Future Work

To achieve a high packet delivery success ratio, this paper presents a novel communication scheme named IOMC for IoT services in LoRa-based LPWAN faced with limited connectivity. IOMC utilizes implicit relay nodes to carry out overhearing operations. IOMC considers the selection of an OH candidate zone and evaluates the average BER and residual energy of the node as the selection strategy of the best OH node, with a backoff-based strategy to eliminate collisions of packets for simultaneous transmission attempts. In general, our analytical results indicated that the use of the overhearing strategy significantly increases the probability of successful transmission, performs best in high node density, and is more resilient to poor RSSI in IoT LoRa networks. IOMC still outperforms two existing schemes, the e-node scheme and the multi-hop scheme, by achieving the highest probability of successful transmission while reducing the number of transmitted packets. In future work, IOMC will be extended to a systematic priority-based adaptive data rate for multi-hop communications in IoT LoRa networks.

## Figures and Tables

**Figure 1 sensors-23-03874-f001:**
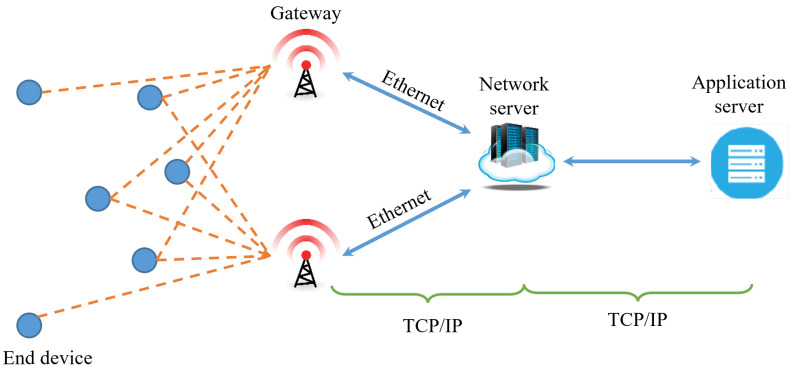
Simple topology for LoRa network.

**Figure 2 sensors-23-03874-f002:**
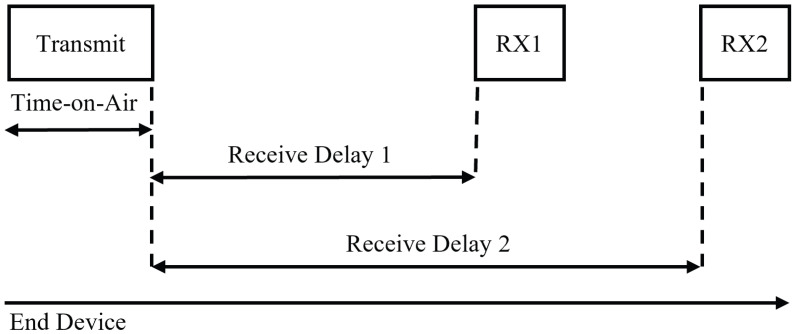
Receive windows in Class A.

**Figure 3 sensors-23-03874-f003:**
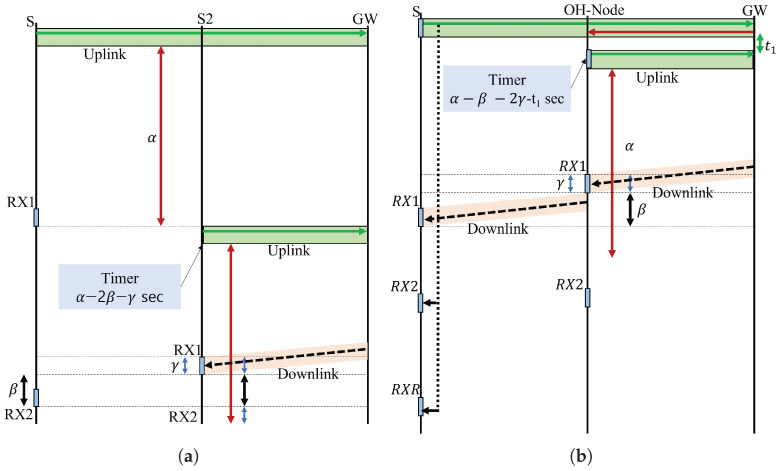
Uplink and downlink message exchange: (**a**) a regular LoRaWAN message transmission; (**b**) IOMC message forwarding mechanism.

**Figure 4 sensors-23-03874-f004:**
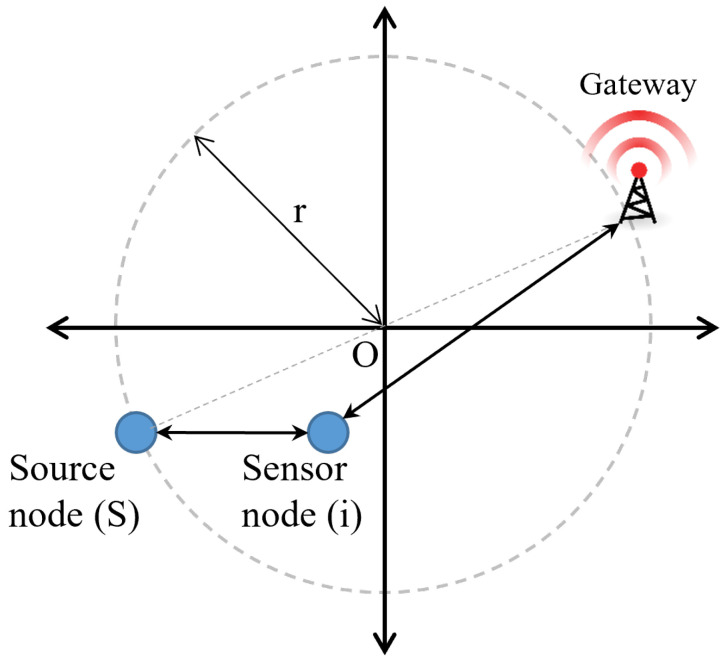
The forwarding zone between a sensor node (S) and the gateway (GW).

**Figure 5 sensors-23-03874-f005:**
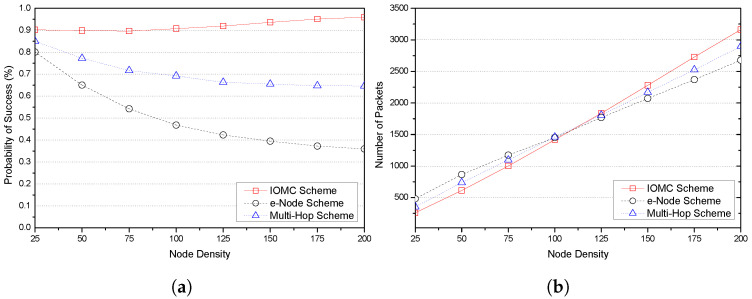
Plots of the performance evaluation metrics: the probability of success and number of packets against node density of various schemes. (**a**) The probability of success as a function of node density. (**b**) The number of packets as a function of node density.

**Figure 6 sensors-23-03874-f006:**
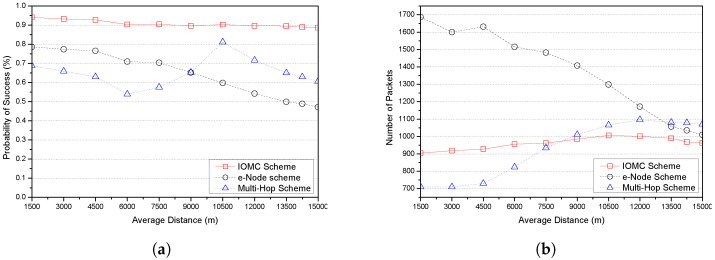
Plot variations in average distance with probability of success and number of packets for various schemes. (**a**) Probability of success vs. average distance. (**b**) Number of packets vs. average distance.

**Figure 7 sensors-23-03874-f007:**
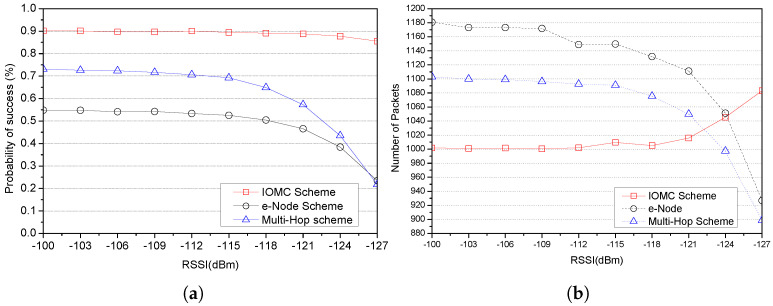
Plot variations in link quality (RSSI) with probability of success and number of packets for various schemes. (**a**) The probability of success as a function of RSSI (dBm). (**b**) Number of packets as a function of RSSI.

**Table 1 sensors-23-03874-t001:** SNR values for different spreading factors.

SF	Sensitivity (dBm)	SNR (dB)
7	−123	−7.5
8	−126	−10
9	−129	−12.5
10	−132	−15
11	−134.5	−17.5
12	−137	−20

**Table 2 sensors-23-03874-t002:** List of main notations.

Symbol	Description
OH(s)	Overhearing node(s)
$d_{SOH}$	Distance from source to OH node
$d_{SGW}$	Distance from source to gateway
$d_{OHGW}$	Distance from OH node to the gateway
BER	Bit error rate
FZ	Forwarding zone
$\gamma_{br}$	SNR of the receiver
$P_{e}\left(S,FN_{i}\right)$	BER from the source to the sensor node in the FN
$E_{single}\left(SF,P_{tx}\right)$	Energy consumption of a single hop
$E_{n-hops}\left(SF,P_{tx}\right)$	Energy consumption of *n* hops
$E_{total\_N_{i}}\left(t\right)$	Total energy consumption
$E_{s}$	Energy consumption of a LoRa node
$E_{re\_N_{i}}\left(t\right)$	Residual energy of a node at time (*t*)
$E_{total\_FN_{i}}\left(t\right)$	Total energy of nodes in the FZ
$E_{total\_re}\left(t\right)$	Total residual energy
$E_{avg\_re}\left(t\right)$	Average residual energy of the nodes in the FZ
$R(OH_{i})$	Rank of a candidate OH node (*i*)
$OH^{*}$	Best OH node
$T_{max}$	Maximum backoff time
$p_{suc(i)}$	Probability of successful transmission

**Table 3 sensors-23-03874-t003:** Simulation parameters.

Parameter	Values
Cell radius (R)	7.5 km
Number of nodes (N)	200
Channel frequency	868 (MHz)
Spreading factor (SF)	7–12
Bandwidth	125, 250, 500 (kHz)
Payload length	30 (Bytes)
Transmission power	14 dBm (25 mW)
Data rate	0.25–5.47 (kbps)
Coding rate	4/5
Simulation time	3600 s
Payload CRC	ON
Interval time	30 s
Limit ToA	30 s
Duty cycle	1%

## Data Availability

Not applicable.

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
