# Peer review of "Implicit Overhearing Node-Based Multi-Hop Communication Scheme in IoT LoRa Networks"

_sensors, 2023, doi:10.3390/s23083874_

Round 1
Reviewer 1 Report
The paper is overall well written and the authors clearly specify the goal of their work. The introduction and the related work section provide a good context and background, which make the discussion easy to follow also for a non-expert in the field.
The proposed methodology, aimed at improving the relay operation in a multi-hop IoT LoRa network environment, is described in detail, and a simulation with NS-3 has been carried out to prove the potential of such an approach. The proposed methodology is pretty intuitive.
The simulation environment section lacks a few details: it is not clear what NS-3 already provides, and what has been implemented by the authors. I guess, for example, that algorithm 1 has been implemented by the authors inside NS-3 simulated IoT nodes. Also, except for the information reported in Table 3, the is no detail about how many times the simulation was carried out. It would be nice if the authors could elaborate a bit more on these two aspects.
As a minor comment, Figure 1 appears first, even if it is referenced for the first much later in the paper.
Author Response
We really appreciate the reviewer's genuine reviews and insightful remarks on the work. These remarks are quite significant and useful as we edit this manuscript.
Please find attached is a detailed response letter.
Best regards.

Reviewer 2 Report
Dear authors,
This is an excellent piece of research. I'd encourage you to improve the following aspects and resubmit.
* use an automatic grammar checker
* Please explain early in the article why have you chosen the name "implicit", I'd say the overhearing is "opportunistic" rather than implicit.
* Please change Figure 2 so that it can properly and harmoniously explain Figure 3. At the moment Figure 2 is confusing and it's got the axis inverted with respect to Figure 3.
* It is not entirely clear how the OH node concludes that the source node has not been able to reach the destination node. Please explain the case in which the data packet does not reach the server and the second case in which the ACK does not reach the source node. How does the opportunistic node make the difference? Please introduce a thorough example in the article.
* The Line 9: in Algorithm 1 is confusing. How is it possible that a pop() function will determine the best node after N iterations? Please clearly define the data structure "candidateset" and explain why pop() does the intended behaviour.
* Please explain with a plot or ranges (time units) the equation (29). It is not clear what kind of order of magnitudes of the T_i and impact of random(1/Tmax) term. Please refer to the time units you expect for T_i.
* Results show that IOMC perform best in high node density and is resilient to poor RSSI. This should be made clear in the abstract and conclusions.
Author Response
We really appreciate the reviewer's genuine reviews and insightful remarks on the work. These remarks are quite significant and useful as we edit this manuscript.
Please find attached is a copy of the response letter.
Best regards.
